# Biomarker Discovery in Rare Malignancies: Development of a miRNA Signature for RDEB-cSCC

**DOI:** 10.3390/cancers15133286

**Published:** 2023-06-22

**Authors:** Roland Zauner, Monika Wimmer, Sabine Atzmueller, Johannes Proell, Norbert Niklas, Michael Ablinger, Manuela Reisenberger, Thomas Lettner, Julia Illmer, Sonja Dorfer, Ulrich Koller, Christina Guttmann-Gruber, Josefina Piñón Hofbauer, Johann W. Bauer, Verena Wally

**Affiliations:** 1EB House Austria, Research Program for Molecular Therapy of Genodermatoses, Department of Dermatology & Allergology, University Hospital of the Paracelsus Medical University, 5020 Salzburg, Austria; mo.wimmer@salk.at (M.W.); m.ablinger@salk.at (M.A.); t.lettner@salk.at (T.L.); ju.illmer@salk.at (J.I.); so.dorfer@crcs.at (S.D.); u.koller@salk.at (U.K.); c.gruber@salk.at (C.G.-G.); j.d.pinon@salk.at (J.P.H.); joh.bauer@salk.at (J.W.B.); 2Center for Medical Research, Medical Faculty, Johannes-Kepler-University, 4020 Linz, Austria; sabine.atzmueller@jku.at (S.A.); johannes.proell@jku.at (J.P.); 3Red Cross Transfusion Service of Upper Austria, 4020 Linz, Austria; norbert.niklas@o.roteskreuz.at; 4Department of Dermatology & Allergology, University Hospital of the Paracelsus Medical University, 5020 Salzburg, Austria; m.reisenberger@salk.at

**Keywords:** epidermolysis bullosa, squamous-cell carcinoma, miRNA, biomarker, exosomes

## Abstract

**Simple Summary:**

Highly aggressive cutaneous squamous-cell carcinomas (cSCCs) are the primary cause of death in patients with the rare skin disease recessive dystrophic epidermolysis bullosa (RDEB). Currently the detection of RDEB-SCCs requires invasive skin biopsies with a high burden for patients and limitations in surveying widespread areas. The aim of our study was to use machine learning as a tool to identify short RNA molecules that are capable of identifying tumors in RDEB patients. As there is only a limited number of RDEB-patients, we included data on patients with a different type of SCCs who, however, show similarity in RNA profiles. This finally facilitated us to nominate sets of short RNAs that can discriminate between tumor and healthy cells.

**Abstract:**

Machine learning has been proven to be a powerful tool in the identification of diagnostic tumor biomarkers but is often impeded in rare cancers due to small patient numbers. In patients suffering from recessive dystrophic epidermolysis bullosa (RDEB), early-in-life development of particularly aggressive cutaneous squamous-cell carcinomas (cSCCs) represents a major threat and timely detection is crucial to facilitate prompt tumor excision. As miRNAs have been shown to hold great potential as liquid biopsy markers, we characterized miRNA signatures derived from cultured primary cells specific for the potential detection of tumors in RDEB patients. To address the limitation in RDEB-sample accessibility, we analyzed the similarity of RDEB miRNA profiles with other tumor entities derived from the Cancer Genome Atlas (TCGA) repository. Due to the similarity in miRNA expression with RDEB-SCC, we used HN-SCC data to train a tumor prediction model. Three models with varying complexity using 33, 10 and 3 miRNAs were derived from the elastic net logistic regression model. The predictive performance of all three models was determined on an independent HN-SCC test dataset (AUC-ROC: 100%, 83% and 96%), as well as on cell-based RDEB miRNA-Seq data (AUC-ROC: 100%, 100% and 91%). In addition, the ability of the models to predict tumor samples based on RDEB exosomes (AUC-ROC: 100%, 93% and 100%) demonstrated the potential feasibility in a clinical setting. Our results support the feasibility of this approach to identify a diagnostic miRNA signature, by exploiting publicly available data and will lay the base for an improvement of early RDEB-SCC detection.

## 1. Introduction

In an orphan disease context, the application of machine learning algorithms to identify tumor-specific biomarkers is often hampered by the limited number of patients and samples. This is also true for the dystrophic subtype of epidermolysis bullosa, which has an estimated prevalence of 6/1,000,000 in the U.S. and Spain [1,2]. Patients suffering from the severe recessive subtype of dystrophic epidermolysis bullosa (RDEB, approximately 11% of all EB cases) are at high risk of developing particularly aggressive cutaneous squamous-cell carcinomas (cSCC) [3]. The cumulative risk of RDEB patients to develop an cSCC by the age of 35 is approximately 68%, reaching 90.1% by 55 years according to the U.S. national EB registry (NEBR), with mortality rates upon SCC development of 87% by the age of 45 [4].

Due to mutations within the COL7A1 gene, RDEB-patients suffer from epithelial blisters and wounds, which frequently chronify due to impaired healing. These chronic wounds are the primary sites for the emergence of cSCCs. The identification of early-stage cSCCs is a major challenge, as they often resemble ulcerations or exuberant granulation tissue, which is very different to cSCCs in otherwise healthy individuals. Understanding the underlying cause for the remarkably aggressive progression of RDEB-cSCCs has been addressed by various groups in the EB research community and the results point towards a complex background influenced by cellular and extracellular changes in patients’ epithelia [5,6,7,8,9].

Currently, detection of RDEB-cSCCs requires meticulous surveillance of non-healing wounds and necessitates invasive skin biopsies with a high burden on patients to confirm malignant lesions; currently, only very limited data is available on less-invasive diagnostic approaches. One report suggests the use of dermatoscopy, which must be performed by experienced, specially trained dermatologists. Another study describes the potential of exploiting an RDEB-cSCC-specific exosomal marker in a xenograft mouse model [10,11]. Thus, the identification of cSCC-specific biomarkers bears great potential for (early) detection and/or treatment monitoring in RDEB-patients. However, given the orphan nature of RDEB, biomarker research faces a dilemma, as it makes it intricately difficult to achieve sample/patient numbers required to produce meaningful and generalizable results. In such settings, the application of machine learning algorithms predestined to discover useful biomarker combinations in omics datasets, which frequently pursue a split train/test data approach, is challenging, as this would require the division of an already small dataset into even smaller ones [12,13]. Thus, attempts to identify similarities between RDEB- and non-RDEB-SCCs should be considered as a possibility to extend and accelerate research by using non-RDEB samples where applicable. This aspect has recently been addressed by Cho et al., who found that RDEB-cSCCs share similarities with head and neck squamous-cell carcinomas (HN-SCC) at the mutational and transcriptional level. In this publication authors investigated and compared 27 RDEB-cSCC, 279 HN-SCC, 38 UV-cSCC [9].

Thus far, no studies exist on potential similarities in miRNA expression between RDEB-cSCC and other tumor entities. Micro-RNAs (miRNAs) have been shown to hold the potential to be used as biomarkers in the context of several malignancies, and their dysregulation has been associated with tumor development and progression at various levels, such as the induction of epithelial-to-mesenchymal transition (EMT), or the conferring of stemness-associated characteristics [14,15]. In general, miRNAs regulate protein expression at the post-transcriptional level in various ways and a classification of distinct miRNAs as onco-, metasta- and tumor suppressor miRs reflects their potency in malignant processes [16]. Tumor-specific expression of miRNAs and their release into the extracellular matrix (ECM) or circulation via exosomes underlines their potential for use as biomarkers. Their stability and relative abundance in particular render them an interesting alternative to biomarkers such as cell-free DNA or circulating tumor cells [17].

In order to develop a miRNA-based diagnostic model for the detection of RDEB-cSCCs, we compared the miRNome of eight different tumors in the cancer genome atlas (TCGA) with our RDEB-cSCC data. On the basis of a high correlation between RDEB-cSCC and HN-SCC miRNA-Seq data, our aim was to use a sufficiently sized, stratified and balanced HN-SCC dataset to train a logistic regression algorithm. A signature of miRNAs with concordant deregulation across both datasets was identified and the predictive performance of the model was tested on independent HN-SCC as well as on RDEB-cSCC data. Further, we were able to demonstrate that these miRNAs can also be found in RDEB-cSCC derived exosomes, thus indicating their potential clinical utility.

## 2. Materials and Methods

### 2.1. Patient Samples and Cell Lines

In this study all experiments were performed with primary cells cultured in defined, serum-free CnT-Prime Epithelial Culture Medium (CELLnTEC, Calgary, Canada) at 37 °C/5% CO_2_ in a humidified incubator. For more detailed information regarding the origin of cell lines and donor description see Table 1.

### 2.2. TCGA Data Retrieval

Curated TCGA miRNA-Seq and clinical data were retrieved using the “curatedTCGAData” package [19]. MiRNAs with low counts were discarded (mean across n samples <10 reads, n: number of samples of smallest experimental group). Library size normalization and variance stabilizing transformation (DESeq2) was performed, referred to as “normalized” data in downstream analysis [20].

### 2.3. miRNA Extraction, Library Preparation and High-Throughput Sequencing

AGO2-bound miRNA was isolated from whole-cell extracts by RNA co-immunoprecipitation (RIP) using the RIPAb + Ago2 validated antibody (Merck-Millipore, Burlington, MA, USA) and the “RNA-Binding Protein Immunoprecipitation kit” (Magna RIP kit, Merck-Millipore, Burlington, MA, USA). The cDNA library from the co-immunoprecipitated small RNA pool was generated using the NEBNext Multiplex Small RNA Library Prep Set for Illumina, San Diego, CA, USA (Set2, New England Biolabs, Ipswich, MA, USA). Fragments containing miRNAs were size selected by gel excision from a 6% TBE-gel. Library quality was assessed using a Bioanalyzer (Agilent 2100, Agilent Technologies, Santa Clara, CA, USA). For the generation of the final sequencing library, three to four samples were equimolar pooled at one time. Molarity for pooling was calculated by fragment size (Agilent) and quantification using PicoGreen (Tecan, Grödig, Austria). The final concentration of the sequencing library was 10 pM with 3% spike-in of PhiX sequencing control. Sequencing was performed on a MiSeq (Illumina) platform according to the manufacturer’s instruction using a 50-cycle v2 MiSeq kit (Illumina).

### 2.4. Exosomal miRNA

MiRNA was extracted from isolated exosomes using miRNeasy serum/plasma kit and miRNeasy tissue/cell advanced kit (Qiagen, Hilden, Germany), according to the manufacturer’s protocol. Sequencing libraries were prepared from isolated exosomal miRNAs using the QIAseq miRNA library kit (Qiagen). Library quality was assessed on a Bioanalyzer (Agilent 2100 or 5200, Agilent Technologies, Santa Clara, CA, USA). Sequencing was performed on 2–3 pooled samples with a final library concentration of 6 pM, spiked with 3% PhiX on an Illumina MiSeq instrument (76-cycles, MiSeq Reagent Kit v3, Illumina, San Diego, CA, USA) or on an Illumina’s NextSeq2000 (100 bp single read mode, Illumina, San Diego, CA, USA) following manufacturer’s instructions.

### 2.5. Exosome Isolation

RDEB-cSCC cell lines (*n* = 6) as well as primary keratinocytes (control KC, *n* = 6) were seeded into T150 flasks and cultured in serum-free CnT-Prime Epithelial Culture Medium (CELLnTEC, Calgary, Canada). The control KC group included exosomes derived from five HC-KC and one RDEB-KC line. The conditioned medium was collected 48 h after cells reached ~80% confluency. A total of 75–120 mL conditioned medium was collected each for RNAseq, Western blot and TEM. Exosomes were isolated by differential centrifugation following a protocol according to standards of the International Society of Extracellular Vesicles (ISEV) [21,22]. First, dead cells were removed by centrifugation at 500× *g* for 5 min at room temperature. The supernatant was further centrifuged at 10,000× *g* (Beckmann JA-14.50 rotor) for 13 min at 4 °C. Extracellular vesicles (EVs) were then harvested by ultracentrifugation of the supernatant at 151,000× *g* for 120 min at 4 °C (Thermo Scientific TH-614 swing bucket rotor, Waltham, MA, USA). Pelleted EVs were washed with PBS, followed by a second round of ultracentrifugation with the same conditions. Finally, the pooled EVs were resuspended in PBS.

### 2.6. Western Blot

Isolated exosomes were lysed in RIPA buffer (Santa Cruz, Dallas, TX, USA) supplemented with 5% β-Mercaptoethanol. After heating samples (95 °C, 5 min) proteins were separated by sodium dodecyl sulfate-polyacrylamide gel electrophoresis with a 4–15% bis-tris gel (Nupage, Invitrogen, Carlsbad, CA, USA) in MOPS running buffer and blotted onto a Hybond-ECL nitrocellulose membrane (Amersham Biosciences, Little Chalfont, UK) in a Towbin transfer buffer. Membranes were blocked in 5% nonfat milk powder in Tris-buffered saline/0.1% Tween 20 (TBST, Merck, Darmstadt, Germany) solution. Primary antibodies mouse anti-CD9 (sc13118, 1:100 diluted, Santa Cruz, Dallas, TX, USA), rabbit anti-α-Actinin (sc15335, 1:1000 diluted, Santa Cruz, Dallas, TX, USA), mouse anti-TSG101 (GTX70255, 1:500 diluted, GeneTex, Irvine, CA, USA) and rabbit anti-GM130 IgG (cs12480S, 1:1000 diluted, Cell Signaling Technology, Cambridge, UK) were incubated overnight at 4 °C. Secondary antibodies used were HRP-labeled Envision+ goat anti-rabbit antibody and goat anti-mouse antibodies (1:1000 diluted in TBST, Dako/Agilent Technologies, Santa Clara, CA, USA). Membranes were washed five times using TBS/0.1% Tween-20 after antibody incubation steps. HRP activity was assessed with the Amersham ECL Select Western blot detection reagent (Amersham Biosciences, Little Chalfont, UK), on a ChemiDoc imager (BioRad, Hercules, CA, USA).

### 2.7. Transmission Electron Microscopy

Transmission electron microscopy (TEM) analysis was performed on a LEO 912AB (Zeiss, Oberkochen, Germany) to visually identify exosomes derived from cultured cells. Briefly, isolated exosomes in PBS suspension were adsorbed on formvar carbon copper grids (200 mesh, Electron Microscopy System, Hatfield, PA, USA) for 30 min. After PBS wash, exosomes were fixed in 1% glutaraldehyde (Electron Microscopy System) for 5 min. After washing steps with sterilized water, samples were stained negative with 10% uranyl-acetate replacement stain (Electron Microscopy System, Hatfield, PA, USA) for 10 min. Air dried samples were observed on TEM under zero-loss energy filtering and an acceleration voltage of 80 kV. Images were recorded via TRS Sharpeye dual speed slow scan CCD camera (Tröndle, Mohrenwies, Germany).

### 2.8. Pre-Processing of RDEB Cell and Exosome miRNA-Seq Data

FastQ files were processed in sRNAtoolbox with default parameter settings [23]. After adapter trimming (NEB-Next or QIA-seq), reads were aligned to the reference genome (GRCh38) with bowtie seed alignment (seed length: 20, minimum read length: 15, allowed mismatches: 2, reads with more than 20 reported alignments were suppressed and alignments with best stratum were reported), alignments were annotated with miRBase v22. Read counts were adjusted for multiple mappings. Precursor expression levels were determined by adding up the counts of reads mapped to the -5p and -3p arms of annotated miRNAs. MiRNAs were normalized for differences in sequencing depth (DESeq2) [20]. Again, MiRNAs with low counts were discarded (mean across n samples <10 reads, n: number of samples of smallest experimental group), the remaining miRNAs were labelled as stable “detected” miRNAs. Batch correction (ComBat_seq, SVA package), library size normalization and regularized log2 transformation (DESeq2) was performed on raw count data, referred to as “normalized” data in downstream analysis [20,24].

### 2.9. Differential miRNA Expression, Correlation, Heatmap, Principal Component Analysis and t-Distributed Stochastic Neighbor Embedding (t-SNE) Analysis

Differentially expressed miRNAs were determined in normalized and adjusted for batch effects and latent surrogate variables (SVA package) RDEB cell miRNA-Seq data using the DESeq2 pipeline [20,25,26]. The default DESeq2 Wald significance test was performed assuming a null hypothesis of no difference between RDEB-cSCC and RDEB-keratinocytes (RDEB-KC). Normalized count data were z-transformed for visualization in a heatmap, an Euclidean distance matrix was used to perform agglomerative hierarchical clustering [27]. Correlation between RDEB-cSCC and TCGA tumor datasets was determined by averaging the normalized counts of overlapping miRNAs between the compared datasets across tumor samples of the respective groups. A linear regression line was fitted, and the Spearman rank correlation (R) calculated [28]. Mean centered, normalized data were used for principal component analysis (PCA) analysis of RDEB-cSCC, RDEB-KC and healthy control (HC)-KC data. For projection of RDEB data into PCA space of HN-SCC samples, both normalized datasets were combined, adjusted for batch effects and SVA and mean centered [25,29]. HN-SCC tumor samples were labeled as members of cluster II based on PCA scores >0 in principal component 2 (Dim2). Normalized, SVA batch adjusted HN-SCC data were used to generate t-SNE plots (Rtsne) with default parameters, the perplexity parameter was set to 20 [30].

### 2.10. Elastic Net Model Training and Performance Evaluation

For model training and testing only those miRNAs were considered that were present in all three datasets: HN-SCC, RDEB-cSCC and exosomes. Based on the stratification of HN-SCC tumor samples in PCA, cluster II samples were removed and only the remaining tumor cluster I (*n* = 277) and normal tissue (*n* = 44) were used for all further downstream analysis. A moderate pre-filter was applied on normalized (DESeq2) miRNA-Seq data to select miRNAs, which showed a minimum of 20% concordant up- or down-regulation (*p*-value < 0.1, Student *t*-test) in RDEB-cSCC versus controls (RDEB-KC and HC-KC), and HN-SCC (cluster I samples) versus HN-normal tissue. The stratified HN-SCC dataset samples were equally split by random sampling into a training and an independent test set using the caret package [31]. Training tumor samples were further randomly downsampled in order to achieve a balanced training set composed of 22 samples each. Normalized (DESeq2) miRNA-Seq count data were z-transformed before training a binomial generalized linear elastic net model (glmnet package). The optimal penalty and mixing parameters were determined using the cv.glmnet function with 10-fold cross validation [32]. Model performance metrics were determined on normalized (DESeq2) and z-transformed stratified HN-SCC test set, as well as on RDEB-cSCC cell and exosomal miRNA-Seq data after using the caret package and receiver operating characteristic (ROC) curves were produced applying the ROCit package [31,33].

### 2.11. qPCR Validation

Starting from whole-cell total RNA samples, cDNA was synthesized in 10 µL reactions using the miRCURY RT Kit (Qiagen, Hilden, Germany) and 10 ng total RNA input. Quantitative (q)PCR reactions were set up as 10 µL reactions using diluted cDNA (1:40), 2× miRCURY SYBR^®^ green master mix (Qiagen, Hilden, Germany) and commercial LNA-enhanced primer assays. Eight out of ten mature miRNAs from the SIG-10 precursor signature (mature let-7a is derived from three different loci) were included based on their dominant arm (-5p/-3p) as well as four different reference RNAs: 5S rRNA, RNU5G, miR-193a-5p and miR-148a-3p (Qiagen, Hilden, Germany). According to the geNORM algorithm 5S rRNA and RNU5G were identified as the most stable reference RNAs and their geometric mean is hereafter used for normalization [34]. Reactions were performed in a 96-well plate format in a Roche LC480 II instrument (Roche, Mannheim, Germany) with the following temperature settings: 95 °C for 10 min, 45 cycles of 95 °C for 10 s and 60 °C for 60 s, followed by melting curve analysis. The second derivative maximum method was used to calculate the cycle of quantification values (Cq-values). For comparison between samples of different experimental groups, 2ΔCq values were calculated as ΔCq = Cq(miRNA target) – Cq(<RNA references>) [35]. List of used LNA primers (Qiagen, Hilden, Germany): miR-7-5p YP00205877, miR-1307-3p P02103132, miR-92b-3p YP00204384, let-7d-5p YP00204124, miR-181a-5p YP00206081, miR-26a-5p YP00206023, let-7a-5p YP00205727, miR-26b-5p YP00204172, hsa-miR-193a-5p YP00204665, hsa-miR-148a-3p YP00205867, 5S rRNA YP00203906 and RNU5G YP00203908.

### 2.12. Data Analysis and Availability

Data analysis was performed and documented in statistical software R (version 3.6.2) with associated packages from Bioconductor and Comprehensive R Archive Network (CRAN). Generated datasets supporting the conclusions of this article are available upon request.

## 3. Results

### 3.1. RNASeq of miRNAs from Primary Keratinocyte and Tumor Cells

In order to discern an RDEB-cSCC associated miRNA profile, cells were isolated from tumor resections as well as from unaffected skin biopsies donated by RDEB patients and healthy donors (Table 1). Sequencing libraries were prepared from AGO2-immunoprecipitated miRNAs isolated from cultured primary cells (RDEB-cSCC: *n* = 6, RDEB-KC: *n* = 5 and HC-KC: *n* = 4) and sequenced after passing a Bioanalyzer quality control. Our choice of specifically employing an AGO2-immunoprecipitation approach was motivated by the fact that (i) AGO2 bound miRNAs are likely to present biological active molecules as AGO2 is part of the miRNA-induced silencing complex, therefore minimizing stochastic noise and (ii) that miRNAs bound to AGO2 or encapsulated in extracellular vesicles are commonly found in bodily fluids, well protected from RNase digestion, which is relevant in the regard of being able to extrapolate cell-based findings to minimally invasive sampling approaches [17]. On average, 5.0 × 10^6^ ± 1.7 × 10^6^ (mean ± s.d.) reads per sample were mapped to the human reference genome (hg38) and annotated with miRBase (v22) mature miRNA identifiers. Reads of both arms (-5p and -3p) were collapsed to their corresponding precursor miRNAs in order to allow compatibility with TCGA datasets. Data were further processed with the DESeq2 package in R to obtain normalized, SVA batch corrected counts and to quantify differential miRNA expression [20,25]. Out of 424 detected miRNAs, 68 were found significantly ≥1.5-fold up- and 95 down-regulated in RDEB-cSCCs (*p*-value ≤ 0.05, Wald significance test, DESeq2) compared to RDEB-KCs. Upon *p*-value adjustment for multiple testing (BH, Benjamini & Hochberg), 51 miRNAs remained significantly (adjusted *p*-value ≤ 0.05) up-regulated and 74 miRNAs down-regulated (Figure 1A) [36].

PCA demonstrated a distinct miRNA expression profile in RDEB-cSCC samples, which separated them clearly from normal, RDEB- and HC-KC samples in the first two dimensions (Dim, Figure 1B). Principal components Dim1 and Dim2 together account for 58% of the total variance in miRNA expression. The observed variance in the data is suggestive of the existence of an inherent signature of deregulated miRNAs within RDEB-cSCCs, which enables their utilization as diagnostic markers.

### 3.2. Similarity of RDEB-cSCC with TCGA Tumor miRNA Profiles

Various machine learning approaches have been successfully deployed to establish binomial classifiers able to differentiate tumors from normal samples based on a variety of molecular features such as miRNAs, proteins and DNA [37,38]. Our intention was to split available RDEB-miRNA data into a set for model training and one for testing (split-data strategy) with randomly assigned samples. This methodology reduces the risk of overfitting, but requires a large amount of data. To surpass the bottleneck of only having access to limited data, we decided to exploit information available from other, potentially similar tumor entities in order to allow the implementation of the split data strategy. Therefore, miRNA-Seq profiles of eight solid tumor types (TCGA code: 01) from different anatomical sites (breast, lung, glandular tissues and mucous linings) were retrieved from TCGA (Figure 2A).

The nomination of these eight tumor entities was based on the availability of a minimum of 40 healthy control samples (TCGA code: 11) in the respective tumor dataset to ensure an appropriate amount of training samples. A hierarchical cluster analysis of TCGA tumor samples showed distinct pan-cancer miRNA expression patterns (Figure 2B). For the assessment of which TCGA tumor profile was most similar to RDEB-cSCC, spearman rank correlation (R) of the mean miRNA expression levels was calculated (Figure 2C). Based on this, HN-SCC showed the highest correlation with RDEB-cSCC (R = 0.75, *p*-value < 2 × 10^−16^), which goes in line with reports showing that the mutational landscape, as well as the transcriptomes of HN-SCC, share similarities to those of RDEB-cSCC. Thus, we considered HN-SCC as suitable complementary data for further downstream analysis [9].

### 3.3. Multivariate Analysis of miRNA Profiles to Stratify HN-SCC Dataset

The HN-SCC TCGA dataset comprises biopsies from different anatomical sites of the aero-digestive mucosa, of tumors in various progression stages, and from patients exposed to multifactorial risks such as smoking, alcohol and human papillomavirus (HPV) infection [39,40]. To identify existing sub-populations with heterogeneous miRNA expression patterns, a PCA analysis was conducted on HN-SCC tumor (*n* = 486) and normal tissue (*n* = 44) data (Figure 3A). Indeed, two tumor sub-populations (clusters I, II) could be distinguished in a PCA plot, as well as in a t-SNE plot (Figure 3B) [30,41]. As PCA uses a weighted linear combination of miRNAs (principal components) to represent variation in miRNA expression, we can infer that miRNAs constituting PCA Dim1 are able to separate both HN-SCC clusters well from HN-normal tissue, whereas PCA Dim2 represents a combination of miRNAs which support the separation of samples from cluster I apart from normal samples. In order to test whether RDEB-cSCC samples would fall into one of the two HN-SCC sub-populations, RDEB samples (6 tumor and 9 normal, including RDEB- and HC-KC) were projected into the HN-SCC PCA space (Figure 3C). A congruent direction of separation between RDEB-cSCC and controls (RDEB-KC and HC-KC) was shared with HN-SCC cluster I (*n* = 277) and respective normal controls. Based on this overlap in PCA we considered HN-SCC cluster I as an appropriate match based on miRNA expression patterns and disregarded cluster II (*n* = 209) samples in subsequent analysis. The removed samples of cluster II in PCA also matched with the majority of samples in cluster II of the t-SNE analysis (Figure 3D). Closer examination of differences in miRNA expression patterns between HN-SCC clusters I and II indicated signatures associated with HPV infection (Appendix A), whereas there were no obvious differences in clinical characteristics such as age or tumor stage (Appendix A). This approach supported a data-driven stratification of the HN-SCC tumor dataset based on miRNA expression patterns to identify a suitable subset of HN-SCC samples to serve in training a tumor prediction model.

### 3.4. Training and Testing of a Logistic Regression Model Using HN-SCC Data

From a repertoire of well-established machine learning classifiers including random forests, support vector machines and boosted trees, we opted to apply an elastic net logistic regression model (Elnet) as it has proven to perform well in the context of tumor prediction and allows an easier and more straightforward interpretation of prediction relevant miRNAs compared to other algorithms [37,38]. It provides a supervised process to learn and regularize regression coefficients *β* (*β*_model_) and parametrize a statistical model able to predict an outcome Y from given data X.

MiRNAs that are present in all used datasets were filtered to pre-select 66 out of 254 detected miRNAs, based on at least 20% concordant (RDEB-cSCC and HN-SCC) deregulation in tumor versus normal samples (*p*-value < 0.1, Student *t*-test on normalized data). The HN-SCC data (*n* = 277 tumor from cluster I and *n* = 44 normal tissue samples) were equally split into two sets by random sampling and labelled as training and test set. Further, the tumor set was downsampled to obtain a balanced training set with equal numbers of tumor (*n* = 22) and normal tissue (*n* = 22) samples. The representativity of the balanced training subset was evaluated based on available clinical data including gender, age, anatomic subsite and pathologic stage as well as common risk factors such as smoking, showing that there was no significant difference in cluster I tumor samples (Appendix A). For training of the prediction model, Elnet provides the option of applying a mix of feature (miRNA) selection by least absolute shrinkage and selection operator (lasso, L1 norm of used coefficients), and ridge penalization (L2 norm of used coefficients, Equation (1)) [32]. In order to tune the model’s mixing parameter *α* (Equation (1)) as well as the penalty parameter *λ* (Equation (2)) and thereby find an optimal bias-variance trade off, a five-fold cross validation (cv.glmnet function in R) was performed on the balanced HN-SCC training dataset to minimize the model error (deviance, Figure 4A) [42]. The cross-validation process determined an optimal mixing parameter (*α*opt = 0.3) which conducts 30% lasso and 70% ridge regression, and controls for highly correlated miRNAs as well as further reduces the amount of miRNAs used in the model. The optimal penalty parameter *λ*opt = 0.0606 was found at *λ*min + one standard deviation. Finally, the Elnet model was fit (glmnet function in R) to the HN-SCC training data. The lasso regularization selected 33 miRNAs (SIG-33 model) with non-zero regression coefficients from the initial 66 features.
(1)$$P\alpha \left(\beta \right)=\left(1-\alpha \right)\frac{1}{2}{\left\|\beta \right\|}_{l2}^{2}+\alpha {\left\|\beta \right\|}_{l1}$$
(2)$$min\left[{\frac{1}{2N}\sum_{i=1}^{N}{\left(y_{i}-\beta_{0}-x_{i}^{T}\beta model\right)}^{2}}^{2}+\lambda P_{\alpha }(\beta )\right]$$

Next, we evaluated the predictive performance of the SIG-33 model with the independent HN-SCC test set. The threshold for labeling a sample as tumor was set to default 0.5, which results in an overall prediction accuracy of 73% (Figure 4B). The SIG-33 model correctly predicted 96/139 (69% sensitivity) tumor and 22/22 (100% specificity) normal tissue samples (Figure 4C). The receiver operating characteristic (ROC) curve illustrates a high discriminative power considering the area under the curve (AUC: 99.98%) of the SIG-33 model to distinguish between the two diagnostic groups (Figure 4D).

### 3.5. Performance Evaluation of the SIG-33 Model with RDEB-cSCC Data

The predictive potential of the elastic net model was next validated on our RDEB-cSCC dataset. The SIG-33 model correctly predicted 6/6 tumors (100% sensitivity) and 9/9 (100% specificity) control (“non-tumor”) keratinocytes (*n* = 5 RDEB-KC, *n* = 4 HC-KC), reaching an overall accuracy of 100% with a threshold set to default 0.5 (Figure 5A). The ROC curve demonstrated a high ability (100% AUC) of the model to differentiate between keratinocytes (RDEB, HC) and RDEB-cSCCs (Figure 5B).

As it has previously been shown that circulating cell-free miRNAs have a high potential as biomarkers due to their frequent dysregulation in cancer, their release into the bloodstream, and their stability mediated, among others, by encapsulation in exosomes, we were interested in whether our prediction model could also be successfully applied to RDEB-cSCC derived exosomal miRNA [43,44]. For this reason we collected the supernatant from primary cells (*n* = 7 RDEB-cSCC, *n* = 6 control keratinocytes including *n* = 5 HC-KC and *n* = 1 RDEB-KC) cultured in serum-free medium and isolated exosomes by differential centrifugation (Figure 5C). The presence of exosomes was confirmed by transmission electron microscopy (TEM) showing vesicles in the expected size range of 50–100 nm with typical cup-shape appearance upon chemical fixation (Figure 5D). Immunoblotting detected exosome-associated CD9, but not golgin subfamily A member 2 (GM130), a marker for cellular contamination, in the exosomal protein lysates. Whole-cell extracts from parental cells served as positive controls (Figure 5E, Appendix A). Total RNA isolated from exosomes showed a peak at the typical size range of small RNAs in a Bioanalyzer electropherogram, whereas RNA from whole-cell extracts showed characteristic ribosomal 18S and 28S peaks (Figure 5F). miRNA-Seq generated an average of 4.4 × 10^4^ ± 4.9 × 10^4^ (mean ± s.d.) reads that were mapped and annotated to miRBase (v22) with 361 detected miRNAs passing low-count threshold.

Finally, the SIG-33 model was tested on the exosome derived, normalized miRNA dataset. It was able to correctly predict 6/7 tumor, (86% sensitivity) and 6/6 (100% specificity) keratinocyte-derived exosome samples with an overall accuracy of 92% at a threshold set to default 0.5 (Figure 5G). The ROC curve demonstrated a high ability (100% AUC) of the model to distinguish between exosomal miRNAs derived from keratinocytes and RDEB-cSCCs, respectively (Figure 5H).

### 3.6. Selection and Performance Evaluation of Models with Reduced Complexity

In total, the Elnet algorithm used 33 features (miRNAs) in its tumor prediction model (Table 2). To determine the possibility of further reducing the model’s complexity by focusing on a subset of features applied by the SIG-33 model, we selected miRNAs by two criteria. First, miRNAs which were found concordantly up- or down-regulated across all three datasets were selected. In a second step, the resulting 29 miRNA subset was further filtered for miRNAs with average read counts across RDEB groups greater than 0.1% of total mapped reads, in order to prioritize strongly expressed miRNAs, which are more likely to be robustly detectable.

As a result, a subset signature designated as SIG-10 was established comprising consistently deregulated miRNAs (up: let-7d, mir-181a-2, mir-7-3, mir-92b and mir-1307; down: mir-26a-2, mir-26b and let-7a-1/2/3) across all three datasets (Figure 6 and Figure 7). All SIG-10 miRNAs were significantly up-/down-regulated in HN-SCC and RDEB-cSCC derived exosomes (adjusted *p*-value < 0.05, DESeq2 Wald *t*-test, BH correction for multiple testing), whereas a higher variation in the expression of signature miRNAs was observed in cultured RDEB-cSCCs with only 2/5 up-, and 1/5 down-regulated miRNAs reaching significance. Despite the wider dispersion in expression of certain miRNAs, employment of the combination of SIG-10 miRNAs enabled the prediction model to compensate for outliers, such as RDEB-SCC63 in mir-1307 counterbalanced by let-7d. Overall, the ability of the less complex SIG-10 model was only moderately impaired in predicting tumor samples compared to SIG-33 (Table 3). Nevertheless, we considered additional downsizing of the SIG-10 model to focus on significantly deregulated miRNAs in order to further improve the robustness of the prediction model. Therefore, miRNAs composing the SIG-10 signature were subjected to qPCR validation to confirm the aberrant expression of the respective mature miRNAs in RDEB cancer cells lines. We chose to examine the mature miRNA expression levels, as sequencing reads mapped predominantly to mature miRNA loci. The dominant strands of precursor miRNAs were determined based on the abundance of miRNA-Seq reads mapped to either the -5p or -3p mature miRNAs (Appendix A). The outcome of the qPCR validation highlighted three miRNAs (SIG-3: miR-7-5p, miR-1307-3p and miR-92b-3p) that were significantly up-regulated (*p*-value < 0.05, non-parametric Wilcox test) in RDEB-cSCCs compared to RDEB-KCs (Figure 8, Appendix A).

In order to demonstrate how well SIG-10 and SIG-3 compete with the full SIG-33 model we evaluated their predictive performance using the AUC-ROC metric as a benchmark. For this purpose, we retrained SIG-10 and SIG-3 on the same balanced HN-SCC training set, which was originally used for the SIG-33 model, and re-assessed their predictive potential on the HN-SCC test set as well as on the RDEB cell and exosome miRNA-Seq data (Table 3). The application of the SIG-10 model reduced the AUC-ROC by 17.3 and 7.1 percentage points in the HN-SCC test set and RDEB-exosome data, respectively, whereas it had no impact on the prediction of RDEB-cSCCs in the cell-based dataset. The use of the least complex SIG-3 signature only led to a modest reduction in model performance (AUC-ROC) by 3.6 and 9.3 percentage points in HN-SCC and RDEB-cSCC, and did not affect the predictive ability in RDEB exosome data. Thus, data suggested that the proposed SIG-3 model is sufficient to detect tumors with high accuracy, specifically based on exosomal miRNAs (Appendix A).

In summary, our machine learning approach revealed altered miRNA expression patterns in HN-SCC and RDEB-cSCC compared to controls, which can be exploited in tumor prediction models. Furthermore, miRNAs characteristic for RDEB-cSCC were detected in respective exosomes, supporting their potential to serve as biomarkers. As a result, three models of varying complexity were derived, and their predictive performance was evaluated.

## 4. Discussion

In this report we demonstrate the feasibility of employing datasets of different tumor entities with a reasonable level of similarity in miRNA expression to train a tumor prediction model. To the best of our knowledge, this pilot study is the first report demonstrating the ability of a combination of miRNAs, which can be found in extracellular vehicles released by tumor cells, in predicting RDEB-cSCCs with high accuracy, thereby establishing the rationale for the potential development of a diagnostic tool.

### 4.1. Biomarkers in RDEB

Despite the fact that EB is an orphan disease, a considerable number of clinical trials are currently being conducted. These include gene-, cell-, and protein-based therapeutics, as well as the use of small molecules and biologics for the treatment of disease complications [45,46,47]. While for the treatment of RDEB-cSCCs the number of clinical trials is increasing, meaningful outcomes are limited due to the particularly aggressive nature of RDEB-cSCCs, the limited response rates of which underpin the urgent need for additional treatment options as well as tools for early diagnosis of RDEB-cSCCs. In order to address the challenge of early detection of RDEB-cSCCs we focused on exploring tumor-specific miRNA expression patterns and tested their potential in predicting RDEB-cSCCs. MiRNAs, together with circulating tumor (ct)DNAs and small non-coding Y-RNAs that have been shown to play a role in DNA replication, RNA stabilization and stress response [48], make up the majority of small RNAs in biofluids and given the high vascularization of RDEB-cSCCs and their high metastatic rate, this disorder seems a good candidate for exploiting such molecules for biomarker research [47,49,50]. The applicability of using miRNA signatures for tumor diagnosis is meanwhile well recognized and their clinical utility has been investigated in various clinical trials [51,52]. Although there is an eminent need for minimally invasive diagnostic tests to support physicians in their effort to detect aggressive RDEB tumors at an early stage, to date there are still only very few studies on potential diagnostic markers. Most of them are at an early observational stage showing a tendency of RDEB-cSCC to present altered levels of certain molecules such as complement associated serine proteases and aberrant expression of a specific cell surface proteoglycan in RDEB-cSCC [53,54]. However, initial studies, including the one presented here, are now elucidating the potential of tumor-associated molecules secreted and protected in microvesicles. Recently it was demonstrated that an exosome-encapsulated RDEB-cSCC RNA biomarker enters the bloodstream, suggesting that an exploitation for diagnostic assays based on less burdensome liquid biopsies could be a feasible approach in future studies [10].

### 4.2. Leveraging Public Data on Malignancies with Potentially Correlating miRNomes for Establishing a Tumor Diagnostic Algorithm

Aiming to generate a well-generalizable model and to facilitate the development of a miR-signature for RDEB-cSCCs, we used publicly available TCGA HN-SCC miRNome data, supplementary to our RDEB data, in order to have sufficient data for independent model training and validation available. Our decision to use HN-SCC data was based on a high degree of similarity to the miRNome profile of RDEB-cSCCs, which has previously also been described for mutation signatures and transcriptomes [9].

Interestingly, in our analysis, it turned out that a certain heterogeneity in miRNA expression existed within the HN-SCC group, and a subset of samples (cluster I) appeared more like RDEB-cSCCs. The major difference between HN-SCC clusters I and II could be attributed to distinct patterns in miRNA expression with a characteristic signature of miRNAs (Appendix A) that have previously been shown to be related to HPV infection in cervical and HN-SCCs [55,56]. In the context of RDEB, the HPV status of patients has been assessed by Purdie et al., who showed that HPV infection is uncommon in RDEB, and also cell lines analyzed in our lab tested negative [57]. Still, a commonality between HN-SCC patients with HPV and individuals living with RDEB is an enrichment in certain mutational signatures, particularly APOBEC [9,58]. However, whether the presence of APOBEC mutational signatures in both RDEB-SCCs and HPV-positive HN-SCCs is linked to the observed similarities in differential miRNA expression is speculative and remains to be investigated.

An alternative approach for the examination of HN-SCC clusters was applied based on clinical features (e.g., tumor stage) or disease associated risk factors (e.g., smoking) provided by TCGA metadata, which did not reveal any significant patterns (Appendix A).

The three proposed models differ in the number of miRNAs used as features to predict tumor samples. Although SIG-33, the model with the highest complexity, outperformed SIG-10 and SIG-3 in their ability to predict tumor samples, caution is warranted, since a higher number of features endows the prediction algorithm with more flexibility, which can instigate an overly optimistic adaptation to training data. In order to circumvent an increased risk of overfitting due to integration of noisy signals, model complexity was reduced [59].

Although using TCGA data from other tumor entities based on the overall similarity of their miRNA expression profile to our specific tumor of interest allows us to increase sample size, this approach also raises the question of the extent to which specificity might be affected. While a definitive test will require collecting further samples of the rare tumor, we addressed this issue indirectly by considering (i) training a model to identify a set of miRNAs capable of distinguishing RDEB-cSCC from HN-SCC samples and (ii) replacing the HN-SCC training dataset with miRNA-seq data from other TCGA tumor classes. A model trained to classify RDEB-cSCC vs. HN-SCC yielded a signature of only three miRNAs (Appendix A, mir-34a, mir-9-1 and mir-9-2), none of which were used in any of the proposed tumor prediction models (SIG-33, SIG-10 and SIG-3), suggesting a high degree of specificity of their miRNA panel to discriminate tumor from non-tumor samples. Further, we also assessed the effects of substituting the HN-SCC training dataset by other TCGA tumor datasets with lower similarity scores (Figure 2C) on the ability to accurately predict RDEB-cSCC tumor samples. Again, the most parsimonious model with the best prediction performance was obtained by training with the TCGA dataset with the highest correlation score when tested on RDEB miRNA-seq dataset (Appendix A).

To assess the effectiveness of applying a feature pre-filter informed by selecting miRNAs with matched regulation, we also considered alternative strategies for incorporating prior knowledge to enhance model specificity. To this end, we considered exploiting available gene expression data derived from cultured primary RDEB-KC/-cSCC cells to conduct functional miRNA target gene set enrichment analysis used as prior knowledge for feature selection (Appendix A). Despite the ensuing performance evaluation producing similar predictive accuracy to SIG-3, the model complexity of SIG-3 was more favorable since only three predictors were used instead of five. In addition, we also tested the option of the glmnet algorithm to integrate sample weights (Appendix A). Therefore, sample weights were derived by determining a similarity distance based on rank correlation between the mean miRNA expression profile of RDEB control/tumor samples with individual HN-SCC samples (cluster I + II). While this concept resulted in high prediction accuracy too, our method—selecting those HN-SCC samples (see PCA, Figure 3C) which clustered closely to RDEB-cSCC samples—outperformed the sample-weighing strategy by offering a significantly less complex model (52 features vs only 33 features in SIG-33).

### 4.3. Signature miRNAs in HN-SCC

In contrast to RDEB-cSCC, HN-SCC is the sixth most common cancer [60]. It is often asymptomatic in its early stages and is therefore only diagnosed at late, already progressed tumor stages. Consequently, the diagnostic and prognostic value of miRNAs in HN-SCC has been explored in numerous studies, in order to advance the development of (early) diagnostic tools [61,62]. In our study we found distinct signature miRNAs that have been previously reported by others, and such that were shown to be functionally related to tumor development and progression, which corroborates the results of this study. For example, Martinez et al. found significantly reduced serum levels of let-7a and miR-26a in liquid biopsies of HN-SCC patients [63], and reduced expression of let-7a was reported in HN-SCC tissue, where this miRNA was linked to tumor immune checkpoint evasion due to accumulation of its target, the programmed cell death ligand 1 (PD-L1) [64,65]. In addition, miR-26a was reported to have tumor-suppressive functions, affecting tumor cell migration by regulating lysyl oxidase like 2 (LOXL2) [66,67]. In a metastudy involving 1685 subjects, Jamali et al. found a relationship between poor prognosis of HN-SCC and expression of miR-181a and let-7d [68]. Elevated levels of miR-92b were found in oral HN-SCC tissue associated with advanced tumor stage and was shown to promote cell proliferation by activation of NF-𝜅B signaling [69]. Finally, increased expression of miR-1307 was reported in laryngeal and oral HN-SCC tissue [70], and elevated expression of miR-7 was associated with poor prognosis in HN-SCC [71,72]. Interestingly, miRNAs that have previously been described to be up-regulated in RDEB-fibroblasts (miR-145) or RDEB-cSCC (miR-10b) do not appear among the top de-regulated miRNAs in our study, which is most likely due to an inhomogeneous regulation across HN- and RDEB-cSCC samples [14,73]. MiR-29c, which was shown to be down-regulated in murine RDEB-skin, is also included in SIG-33 [74].

### 4.4. Limitations and Future Opportunities

As the major focus of this study was on finding an appropriate set of miRNAs suitable to detect RDEB-cSCCs, we used a stratified, highly similar subset of HN-SCC samples from TCGA as a surrogate to extend sparsely available RDEB-cSCC samples. In order to further optimize and generalize the proposed model for a specific use in HN-SCCs, a retraining on a set including the afore disregarded cluster II samples would be advisable, as the predictive accuracy of the currently proposed model decreases by approximately five percentage points if applied to HN-SCC cluster II (Appendix A).

Cumulating evidence suggests the presence of miRNAs in cell-derived exosomes and their possible use as biomarkers [75,76]. In relation to that, we were able to confirm for the first time the existence of tumor associated signature miRNAs in exosomes derived from malignant and non-malignant primary RDEB keratinocytes. Of note, we observed no significant difference in the expression of cell-derived signature miRNAs between RDEB- and HC-KC, enabling a clear distinction of tumor from non-tumor also in exosome-derived signature miRNA. Due to limitations in the availability and expansion capacity of primary RDEB-KCs, which is needed for exosome isolation, we were only able to include one RDEB-KC line in these analyses. This represents a limitation of the study, but nevertheless demonstrated that the exosomal signature miRNA levels from this RDEB-KC line grouped nicely to those obtained from HC-KCs. Although the use of data derived from primary cultured cells isolated from biopsies of tumor resections limits the clinical generalizability of our observations, this study provides compelling basic evidence to support validation in a clinical setting.

## 5. Conclusions

The discovery of a tumor characteristic miRNA signature contributes to a potential future clinical application of biomarker-based diagnostic tools for RDEB-cSCCs. In addition, we present a possible strategy to circumvent limitations in sample size and patient numbers, by exploiting public data from HN-SCCs, based on similarities in miRNA expression patterns. Although the clinical applicability of our signature remains to be evaluated, we demonstrated that a combination of three miRNAs can not only distinguish tumor cells from non-tumor samples, but that this signature can also be found in microvesicles released by tumor cells, establishing an avenue for future development of minimally invasive diagnostic tests.

## Figures and Tables

**Figure 1 cancers-15-03286-f001:**
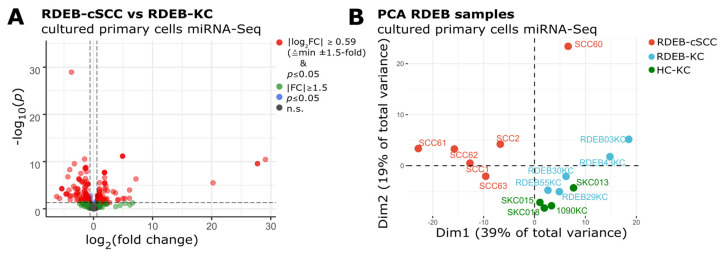
(**A**) Volcano plot shows differentially expressed miRNAs in RDEB−cSCC compared to RDEB−KC (FC: fold change, *p*: Wald test for significance DESeq2, n.s.: non-significant *p* > 0.05). (**B**) PCA plot indicates that the RDEB−cSCC-specific miRNA expression profile enables their separation from RDEB−KC and HC−KC samples.

**Figure 2 cancers-15-03286-f002:**
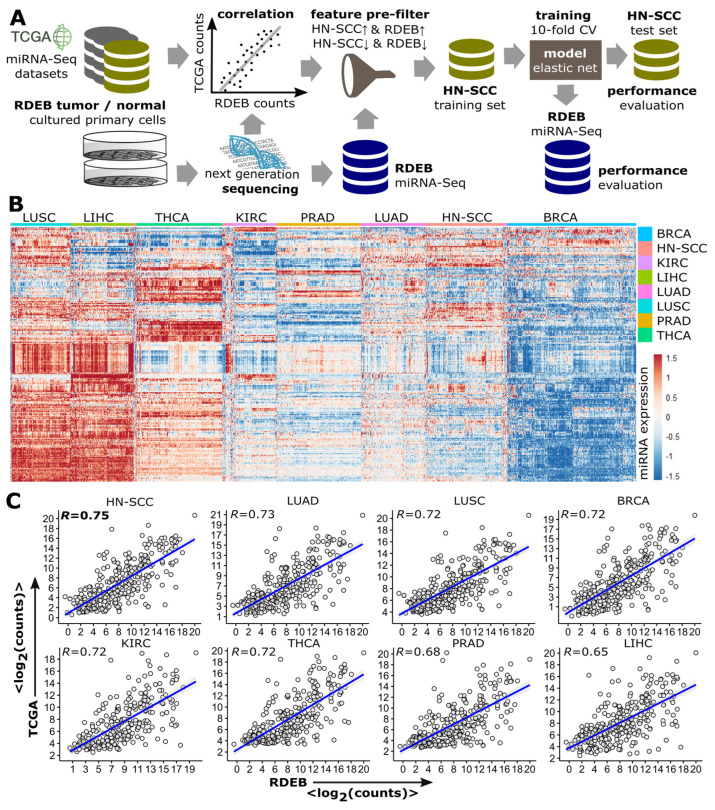
Similarity between RDEB−cSCC and solid TCGA tumors. (**A**) Data analysis workflow. TCGA miRNA−Seq tumor datasets were correlated with RDEB−cSCC sequencing data from cultured primary cells to identify tumor types with miRNA expression patterns similar to RDEB−cSCC. The most similar TCGA data is randomly split and a part of it is used—after applying a feature pre-filter by selecting concordantly regulated miRNAs when comparing expression in HN-SCC and RDEB samples—to train an elastic net logistic regression model, its performance is subsequently evaluated on independent TCGA and RDEB test sets. (**B**) Heatmap of miRNA expression levels (rows) of tumor samples (columns) from eight different TCGA tumor types shows distinct and tumor type associated expression patterns (red indicates an increased and blue a decreased miRNA expression, normalized and z-transformed read counts). (**C**) Scatterplots of mean log2 normalized (DEseq2) miRNA-Seq counts (each dot represents one miRNA) of TCGA tumor and RDEB−cSCC samples. (R spearman rank correlation, *p* < 2.2 × 10^−16^ for all 8 comparisons). (BRCA: breast invasive carcinoma, KIRC: Kidney renal clear cell carcinoma, LUAD: lung adenocarcinoma, LIHC: liver hepatocellular carcinoma, LUSC: lung squamous-cell carcinoma, PRAD: prostate adenocarcinoma and THCA: Thyroid carcinoma).

**Figure 3 cancers-15-03286-f003:**
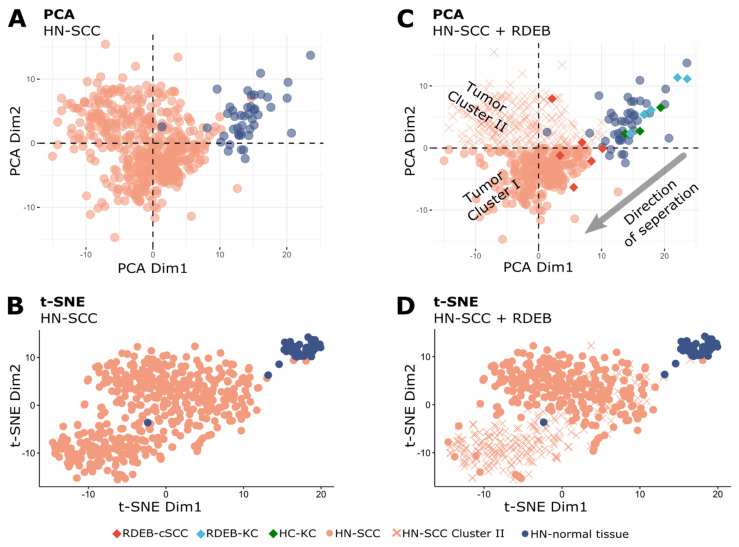
Multivariate analysis of HN−SCC. (**A**) PCA plot of the first two principal components of normalized (DESeq2) miRNA-Seq data of TCGA/HN−SCC shows a distinct cluster for normal tissue samples, as well as two tumor clusters (each dot represents one sample). (**B**) t−SNE plot of HN−SCC data also presents three distinct clusters. (**C**) Plot shows RDEB samples projected into HN−SCC PCA space with separation of RDEB−cSCC from RDEB−KC and HC−KC samples along the direction of a separation of HN−SCC tumor cluster I and HN-normal tissue samples (x: HN−SCC tumor cluster II samples). (**D**) HN−SCC samples from tumor cluster II (PCA plot) are associated with one of the two distinct tumor clusters in the t−SNE plot.

**Figure 4 cancers-15-03286-f004:**
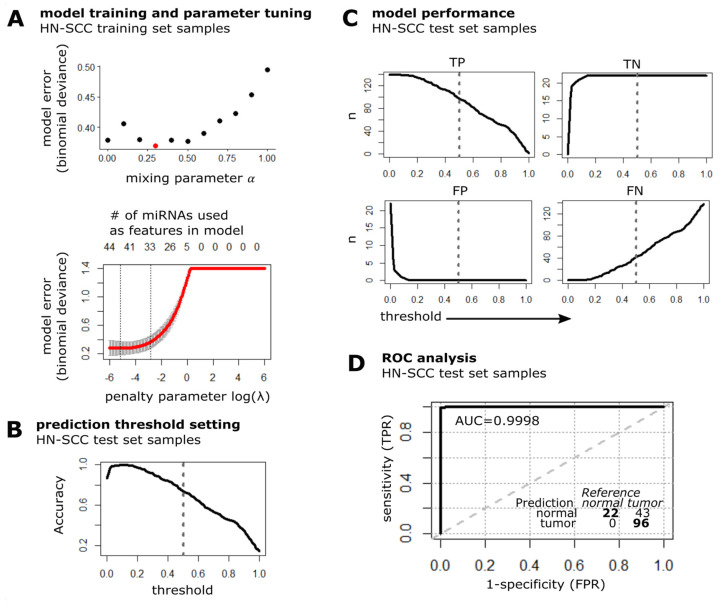
Model training and performance test in cluster I HN-SCC. (**A**) Plot illustrates the prediction error of fitting the Elnet model to the HN-SCC training dataset with different values for the mixing parameter α (red dot indicates *α*opt = 0.3) and penalty parameter *λ* (data represented by mean ± s.d.). The optimal penalty parameter *λ*opt = 0.0606 (right dashed line) was chosen such that the error falls within one standard error of *λ*min = 0.0058 (left dashed line). (**B**) Accuracy of SIG-33 model when tested on HN-SCC test set samples is plotted over a range of threshold values, dashed line indicates default threshold of 0.5. (**C**) Detection metrics of the Elnet SIG-33 model upon prediction of HN-SCC test data (n: number of samples classified as either TP: true positive, TN: true negative, FP: false positive and FN: false negative). Dashed line represents threshold set to 0.5. (**D**) ROC curve and confusion matrix of SIG-33 model performance evaluation on HN-SCC test dataset. TPR: true positive rate, FPR: false positive rate, AUC: area under curve, bold line: empirical ROC curve and dashed: chance line.

**Figure 5 cancers-15-03286-f005:**
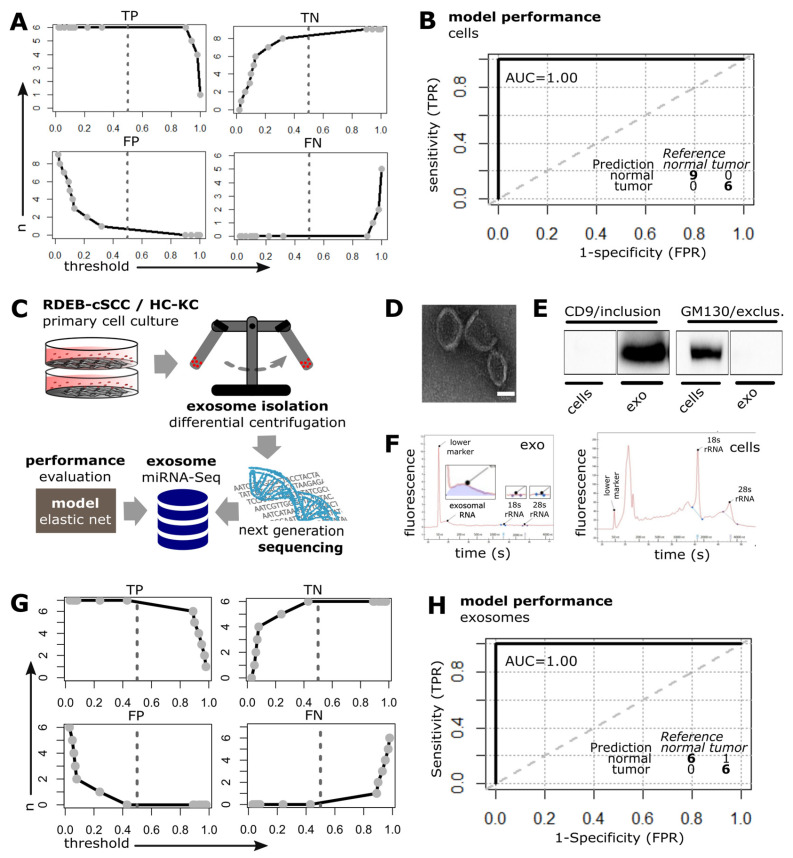
SIG-33 model performance in cultured cells and exosomes. (**A**) Elnet model was tested on cell-based RDEB miRNA-Seq data including RDEB-cSCC (*n* = 6), HC (*n* = 4)/RDEB-KC (*n* = 5) (*n*: amount of samples classified as either: TP: true positive, TN: true negative, FP: false positive, FN: false negative compared to the labeled reference, dashed line represents default threshold set to 0.5 and bold lines: interpolation between point estimates). (**B**) ROC curve illustrates the model’s predictive performance on RDEB-cSCC data. Inset summarizes the prediction results at threshold set to default 0.5. (TPR: true positive rate, FPR: false positive rate, AUC: area under curve, bold line: empirical ROC curve and dashed: chance line). (**C**) Workflow for generation of exosomal miRNA-Seq data. (**D**) TEM of exosomes isolated from conditioned medium of cultured primary cells (scale bar: 50 nm). (**E**) Western blot bands show positive signal for exosome marker CD9 and negative signal for impurity related Golgi marker GM130 in exosomal protein lysates (RDEB-SCC1), whole-cell lysates (RDEB-SCC1) were used as control (complete Western blot image is shown in Appendix A). (**F**) Electropherograms show characteristic profiles for RNA isolated from exosomes ((**left panel**), low to non-existent 18S/29S rRNA peaks, see insets) as well as RNA extracts from whole cells (**right panel**). (**G**,**H**) Threshold and ROC plots: Elnet model tested on sequencing data generated from exosomal miRNAs including RDEB-cSCC (*n* = 7), HC (*n* = 5)/RDEB (*n* = 1)-KC.

**Figure 6 cancers-15-03286-f006:**
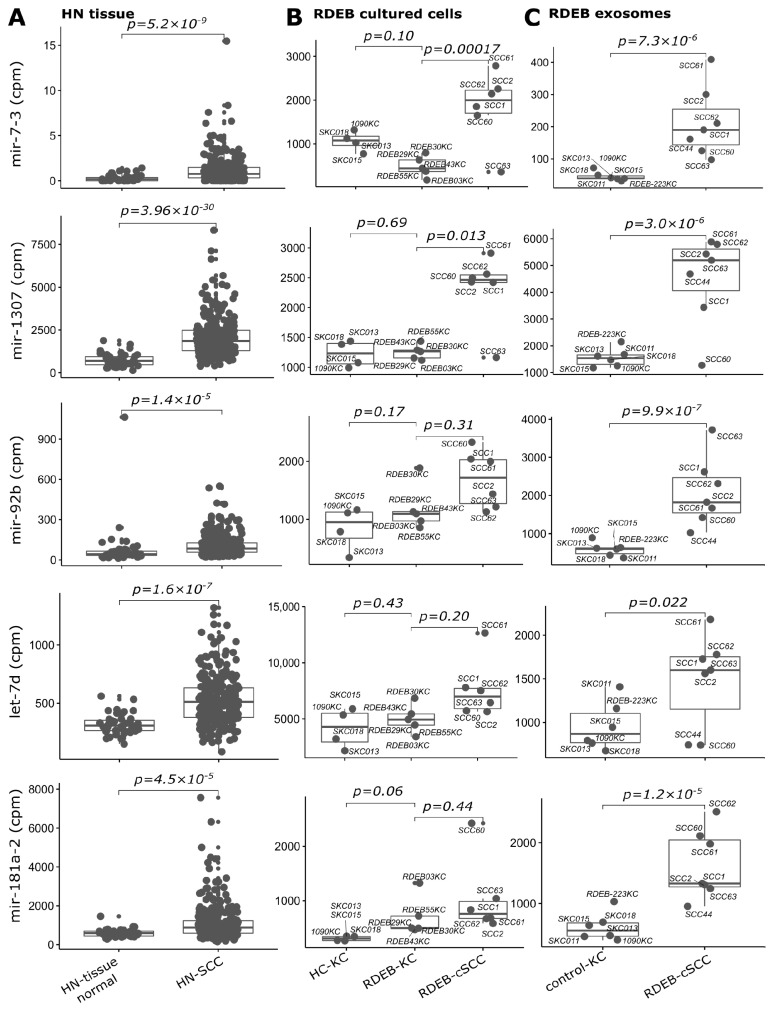
SIG-10 miRNAs up-regulated in tumors. Boxplots illustrate sequencing counts normalized by library size (mapped reads, cpm: counts per million) of five miRNAs used in the prediction model, which show concordant up-regulation in (**A**) HN tissue: HN-SCC (cluster I, test set), (**B**) RDEB cultured cells: *n* = 6 RDEB-cSCC, *n* = 5 RDEB-KC and *n* = 4 HC-KC) and (**C**) RDEB exosomes: *n* = 7 RDEB-cSCC, *n* = 1 RDEB-KC and *n* = 5 HC-KC data, compared to respective controls (HN tissue: non-tumor tissue samples, RDEB cultured cells: HC/RDEB-KC samples grouped separately and RDEB exosomes: combined HC/RDEB-KC samples). (*p*-value: significance, DESeq2 Wald test, Benjamini & Hochberg multiple testing adjusted). Each dot represents one sample, small dots indicate outliers. For exosomes, RDEB- and HC-KCs were combined into “control-KC” group.

**Figure 7 cancers-15-03286-f007:**
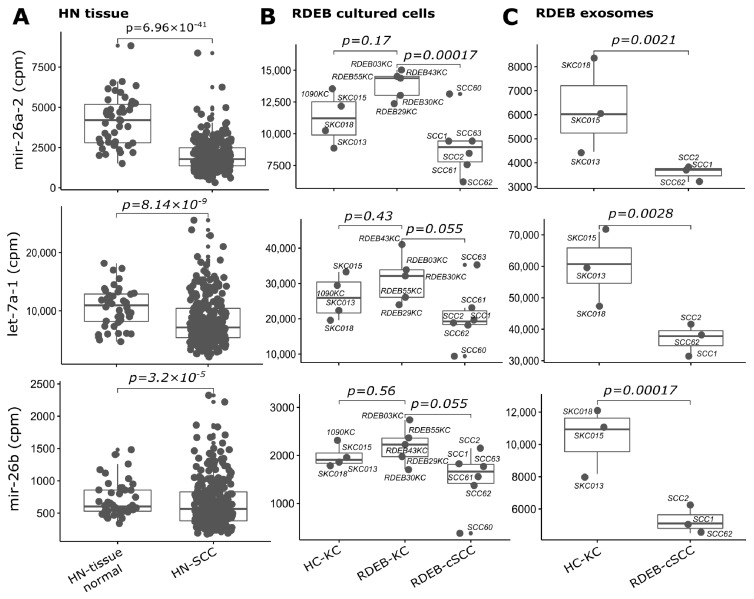
SIG-10 miRNAs down-regulated in tumors. Boxplots illustrate sequencing counts normalized by library size (mapped reads, cpm: counts per million) of three miRNAs (let-7a-1 is representative for paralogues let-7a-2 and let-7a-3) used in the prediction model, which show concordant down-regulation in (**A**) HN tissue: HN-SCC (cluster I, test set), (**B**) RDEB cultured cells: *n* = 6 RDEB-cSCC, *n* = 5 RDEB-KC, *n* = 4 HC-KC) and (**C**) RDEB exosomes: *n* = 7 RDEB-cSCC, *n* = 1 RDEB-KC, *n* = 5 HC-KC data, compared to respective controls (HN tissue: non-tumor tissue samples, RDEB cultured cells: HC/RDEB-KC samples grouped separately, RDEB exosomes: combined HC/RDEB-KC samples). (*p*-value: significance, DESeq2 Wald test, Benjamini & Hochberg multiple testing adjusted). Each dot represents one sample, small dots indicate outliers. For exosomes, RDEB- and HC-KCs were combined into “control-KC” group.

**Figure 8 cancers-15-03286-f008:**
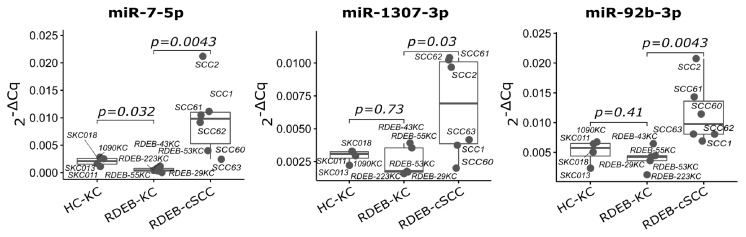
qPCR validation of SIG-3 miRNAs in RDEB tumor cells. Boxplots show relative expression of three significant mature miRNAs used in the SIG-3 prediction model. TaqMan qPCR was performed on cultured RDEB cells. (*p*-value: significance, non-parametric unpaired Wilcox test, mean of 5S RNA and RNU5G Cq was used as reference). Each dot represents one sample.

**Table 1 cancers-15-03286-t001:** List of primary keratinocytes and RDEB-cSCCs.

	Age (Yrs)	Sex	COL7A1 Mutation	Localization	Sample Type	Ref.
1090KC	23	m	healthy control	arm	HC-KC	-
SKC011	28	f	healthy control	abdominal	HC-KC	-
SKC013	35	f	healthy control	abdominal	HC-KC	-
SKC015	36	f	healthy control	abdominal	HC-KC	-
SKC018	31	m	healthy control	abdominal	HC-KC	-
RDEB-03KC	18	m	c.425A > G/c.5440G > T	upper arm	RDEB-KC	-
RDEB-29KC	2	m	c.425A > G/c.520G > A	foreskin	RDEB-KC	-
RDEB-30KC	2	m	c.425A > G/c.520G > A	foreskin	RDEB-KC	-
RDEB-43KC	17	f	c.4027C > T/c.425A > G	inner thigh	RDEB-KC	-
RDEB-55KC	21	m	c.976 + 4A > C homozygous	upper thigh	RDEB-KC	-
RDEB-223KC	2	f	c.425A > G/c.425A > G	upper arm	RDEB-KC	-
RDEB-SCC1	32	f	c.8244dupC homozygous	SCC shoulder	RDEB-cSCC	SCCRDEB4 [18]
RDEB-SCC2	54	m	c.3832-1 G > A/undetermined	SCC forearm	RDEB-cSCC	RDEBSCC_02 [9]
RDEB-SCC44	31	m	c.8440C > T homozygous	SCC foot	RDEB-cSCC	RDEBSCC_06 [9]
RDEB-SCC60	27	m	c.425 A > G homozygous	SCC elbow	RDEB-cSCC metastasis	RDEBSCC_27 [9]
RDEB-SCC61	24	m	c.6527dupC/c.5532 + 1G > T	SCC back	RDEB-cSCC	RDEBSCC_29 [9]
RDEB-SCC62	29	f	c.682 + 1G > A/c.7474C > T	SCC lower arm	RDEB-cSCC	RDEBSCC_09 [9]
RDEB-SCC63	50	m	c.5532 + 1G > T/c.3264_5293del	SCC back	RDEB-cSCC	RDEBSCC_30 [9]

**Table 2 cancers-15-03286-t002:** List of 33 miRNAs selected by the Elnet regression SIG-33 model.

miRNA	*β* _model_	HN-SCC	RDEB-cSCC	Exo	miRNA	*β* _model_	HN-SCC	RDEB-cSCC	Exo
**let-7d**	**+0.479**	**up**	**up**	**up**	mir-654	−0.1390	down	down	up
mir-181a-1	+0.374	up	up	up	mir-379	−0.1350	down	down	down
mir-29c	−0.322	down	down	down	**mir-7-3**	**+0.1300**	**up**	**up**	**up**
let-7c	−0.298	down	down	up	mir-218-2	−0.1260	down	down	down
mir-181b-1	+0.288	up	up	up	mir-411	−0.0979	down	down	down
mir-1306	+0.256	up	up	up	**mir-92b**	**+0.0853**	**up**	**up**	**up**
**mir-26a-2**	**−0.244**	**down**	**down**	**down**	mir-483	+0.0848	up	up	up
mir-125b-2	−0.241	down	down	up	mir-362	−0.0820	up	down	up
mir-301b	+0.226	up	up	up	mir-1292	+0.0445	up	up	up
mir-301a	+0.215	up	up	up	mir-345	+0.0433	up	up	up
mir-877	+0.215	up	up	up	mir-1910	+0.0361	up	up	up
mir-660	−0.179	down	down	down	**let-7a-3**	**−0.0281**	**down**	**down**	**down**
mir-589	+0.171	up	up	up	**let-7a-1**	**−0.0274**	**down**	**down**	**down**
mir-337	−0.164	down	down	down	**let-7a-2**	**−0.0271**	**down**	**down**	**down**
**mir-26b**	**−0.162**	**down**	**down**	**down**	mir-136	−0.0218	down	down	down
mir-502	−0.162	down	down	down	**mir-1307**	**+0.0141**	**up**	**up**	**up**
**mir-181a-2**	**+0.141**	**up**	**up**	**up**					

*β*_model_: Elnet regularized regression coefficients (relative weight for each miRNA). Up/down indicates the average up- or down-regulation in “HN-SCC versus HN-normal tissue” (HN-SCC), “RDEB-cSCC versus RDEB-KC” (RDEB-cSCC), and “RDEB-cSCC exosomes versus HC-KC exosomes” (Exo). Strongly expressed miRNAs with average read counts in RDEB-cSCC or RDEB-KC > 0.1% of total reads per sample are marked bold.

**Table 3 cancers-15-03286-t003:** Model performance comparison.

	SIG-33	SIG-10	SIG-3
HN-SCC	99.98	82.73	96.37
RDEB-cSCC	100.00	100.00	90.74
RDEB exosomes	100.00	92.86	100.00

Performance of full SIG-33 as well as sparse SIG-10 and SIG-3 tumor prediction models is presented by % area under the receiver operating curve (AUC-ROC). HN-SCC: cluster I test set.

## Data Availability

The data that support the findings of this study are available upon request.

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
