# Peer review of "Biomarker Discovery in Rare Malignancies: Development of a miRNA Signature for RDEB-cSCC"

_cancers, 2023, doi:10.3390/cancers15133286_

Round 1
Reviewer 1 Report
In this study, Zauner et al. propose a novel approach employing an elastic net generalized linear model to predict miRNA signatures for RDEB-cSCC. To address the limitation of sample accessibility, the authors analyze miRNA expression profiles of HN-SCC from the TCGA repository to identify similarities and find an appropriate match. The results demonstrate the feasibility of this approach in identifying a diagnostic miRNA signature, potentially improving early RDEB-SCC detection. I believe the manuscript can be further enhanced in the following areas:
The idea of combining external datasets from different subjects to increase sample accessibility is intriguing. However, this approach may also result in less specificity for the model trained. It is essential for the authors to demonstrate that such factors are properly addressed. A possible solution is to apply a similar model as a binary classification for RDEB-cSCC/non-RDEB-cSCC samples, identify potential predictors, and carefully examine these predictors to ensure their validity.
The authors should consider whether it is possible to incorporate any form of prior knowledge for parameter selection in the regression model. For instance, miRNet provides networks for miRNA functional analysis. Prior information on the interactions between miRNAs could be valuable in deriving important markers for the prediction model. Incorporating such knowledge may strengthen the model and further enhance its performance.
By addressing these points, the manuscript will provide a more comprehensive understanding of the proposed approach, its limitations, and potential implications for the field of miRNA signature prediction and early detection of RDEB-cSCC.
Author Response
- The idea of combining external datasets from different subjects to increase sample accessibility is intriguing. However, this approach may also result in less specificity for the model trained. It is essential for the authors to demonstrate that such factors are properly addressed. A possible solution is to apply a similar model as a binary classification for RDEB-cSCC/non-RDEB-cSCC samples, identify potential predictors, and carefully examine these predictors to ensure their validity.
Response 1.1: We thank Reviewer#1 for advising to consider further evaluation our approach using external datasets to increase sample size, in particular how different choices effect the specificity of the model. We appreciate the specific suggestion of interrogating a binary classification model and find this idea very useful. Therefore, we have expanded our analysis and conducted an additional round of model training to learn differences between RDEB-cSCC and non-RDEB-cSCC (HN-SCC) and found a set of three miRNAs. This “inter-tumor” signature has not shown any overlap with our current “tumor vs normal” SIG-33, SIG-10 and SIG-3 signature panel. In addition, we also tested whether using TCGA datasets other than HN-SCC would result in similar or better models predicting RDEB-cSCC. Again, we could demonstrate that our initial approach using the most similar TCGA tumor set in terms of their miRNA expression profile to train the prediction model outperformed using less similar (correlation analysis, Fig.2C) TCGA tumor datasets. We have included respective paragraphs in the discussion section, see line# 638-654 and added data in the additional file – Table S3, S4.
- The authors should consider whether it is possible to incorporate any form of prior knowledge for parameter selection in the regression model. For instance, miRNet provides networks for miRNA functional analysis. Prior information on the interactions between miRNAs could be valuable in deriving important markers for the prediction model. Incorporating such knowledge may strengthen the model and further enhance its performance.
Response 1.2: We agree with Reviewer#1 that it is useful to consider alternative approaches and specifically address the possibility of integrating prior existing knowledge. To this end, we additionally highlighted the fact that we were already using “prior” knowledge during the feature selection step (see updated Fig. 2A, filter step now also emphasized in workflow) – where we applied a “mild” pre-filter to select miRNAs with congruent up/downregulation in RDEB-cSCC and the training dataset (HN-SCC) in order to prioritize miRNAs which should provide relevant information to the classification model. Furthermore, we are now providing additional data on different strategies to incorporate prior knowledge and tested whether this would allow us to further enhance model specificity and performance. Inspired by the reviewer’s suggestions we exploited available gene expression data derived from cultured primary RDEB tumor cells/keratinocytes using geneset enrichment analysis to predict miRNA activity in those cells, which were used as prior knowledge to narrow down “informative” miRNAs during feature selection. Another test was evaluating the use of sample weights – a feature offered by the glmnet algorithm. Both cases demonstrate that the use of prior knowledge actually improves model performance, although none of the methods was able to outcompete our current chosen configuration. See additional text passage in discussion section line #655-#670 and tables and figures in additional file - Table S5, Fig. S6, S7.
By addressing these points, the manuscript will provide a more comprehensive understanding of the proposed approach, its limitations, and potential implications for the field of miRNA signature prediction and early detection of RDEB-cSCC.
Reviewer 2 Report
The Review submitted by Zauner et al. entitled Biomarker discovery in rare malignancies: development of a 2 miRNA signature for RDEB-cSCC characterized a microRNA signature for the potential detection of tumors in RDEB patients. The objective of a study is important and the approach that authors used is interesting. The paper is clearly written, and well-organized. I only have minor issues:
1. I was confused about the compared groups in the paper. As I understood, the aim was to distinguish between RDEB-cSCC and RDEB-KC, but part of the comparisons was between HC-KC and RDEB-cSCC, why not RDEB-KC? The authors should more clearly write which samples were taken for a certain comparison, examples:
- Page 9, line 349, which controls do the authors refer to?
- Page 14, line 495-496, please specify 3 datasets, up- or downregulated compared to what?
- Page 17, line 538, Does control KCs mean HC-KC or RDEB-KC?
2. The aim of the study was to “…develop a miRNA-based diagnostic model for the detection of RDEB- cSCCs. The authors should more clearly and straightforward write what is the outcome and conclusion of the study. Do the authors recommend to use the 3 identified microRNAs to distinguish between RDEB-KC and RDEB-cSCC and diagnose RDEB-cSCC? Is it better to use exosomes or cellular miRNAs?
3. The authors should also specify why do they perform Ago2-IP before isolating miRNAs for sequencing. They should also rephase the name cellular miRNAs to AGO-bound miRNAs, since phrase “cellular miRNAs” suggests simple miRNA isolation from cells.
4. Fig 5F, RNA from cell extracts looks as if small RNAs were enriched, and in exosomes too little RNA compared to marker was added.
Author Response
The Review submitted by Zauner et al. entitled Biomarker discovery in rare malignancies: development of a 2 miRNA signature for RDEB-cSCC characterized a microRNA signature for the potential detection of tumors in RDEB patients. The objective of a study is important and the approach that authors used is interesting. The paper is clearly written, and well-organized. I only have minor issues:
- I was confused about the compared groups in the paper. As I understood, the aim was to distinguish between RDEB-cSCC and RDEB-KC, but part of the comparisons was between HC-KC and RDEB-cSCC, why not RDEB-KC? The authors should more clearly write which samples were taken for a certain comparison, examples:
- Page 9, line 349, which controls do the authors refer to?
- Page 14, line 495-496, please specify 3 datasets, up- or downregulated compared to what?
- Page 17, line 538, Does control KCs mean HC-KC or RDEB-KC?
Response 2.1: We thank Reviewer#2 for pointing out that the use of the term “control” group might be to unspecific. Therefore, explicit and detailed info is now provided where appropriate; see line #358-359, #434, #437, #443-444, #466, #480, #501-506, #511-517.
The reason for supplementing RDEB-KC samples with HC-KC, which were summarized under the term “controls”, was to increase sample size. RDEB-KC, and in specific primary RDEB-KC as used in this study are rare and hard to obtain as tissue biopsies present a high burden to patients alongside with complications such as high risk of infections. In a previous study (Wimmer et al 2020) we found no striking difference in the miRNA profiles of RDEB-KC and HC-KC and therefore argue that mixing these two groups in a combined non-tumor control group is appropriate in this case. Nevertheless, we clearly labelled the sample origin in the boxplots to allow readers to draw their own conclusion about the abundance levels of specific miRNAs in respect to HC- vs RDEB-KC.
- The aim of the study was to “…develop a miRNA-based diagnostic model for the detection of RDEB- cSCCs. The authors should more clearly and straightforward write what is the outcome and conclusion of the study. Do the authors recommend to use the 3 identified microRNAs to distinguish between RDEB-KC and RDEB-cSCC and diagnose RDEB-cSCC? Is it better to use exosomes or cellular miRNAs?
Response 2.2: We expanded our conclusion (see lines#724-728) not only highlighting our strategy to circumvent limitations in sample size by learning from similar data, but also outlining that our SIG-3 miRNA signature which proved similarly effective in cells and exosome-derived data provides an opportunity to test this prediction model in future studies. The fact that exosomes provide the same information as their parental cells provides us with a rationale to screen minimally invasive patient samples for our SIG-3 signature.
- The authors should also specify why do they perform Ago2-IP before isolating miRNAs for sequencing. They should also rephase the name cellular miRNAs to AGO-bound miRNAs, since phrase “cellular miRNAs” suggests simple miRNA isolation from cells.
Response 2.3: We agree with Reviewer#2 that it is of interest to the reader to provide the rationale for using AGO2-IP isolated miRNAs. Our motivation for using AGO2-IP is now explicitly outlined in section 3.1 line#273-282 The term “cellular miRNAs” was removed from the method section (2.3).
- Fig 5F, RNA from cell extracts looks as if small RNAs were enriched, and in exosomes too little RNA compared to marker was added.
Response 2.4: We have adapted Fig.5F to highlight (magnify) the region of interest showing non existent 18S/28S peaks indicative of small RNA enrichment in exosomes. Figure legend was adapted, line#478. As indicated by the reviewer, the NEBNext seq-lib preparation protocol includes a size-based gel excision step of amplicons, thus a certain enrichment of small RNAs is to be expected.
Round 2
Reviewer 1 Report
I commend the authors for the substantial improvements made in the revised manuscript. The inclusion of analysis on dataset characteristics enhances the study's scientific rigor. The discussion on the coherence between prior knowledge and experimental data provides a comprehensive analysis of the model predictors. Considering these improvements, I recommend accepting the manuscript for publication in its present form.